# Micromechanism of Cold Deformation of Two-Phase Polycrystalline Ti–Al Alloy with Void

**DOI:** 10.3390/ma12010184

**Published:** 2019-01-07

**Authors:** Ruicheng Feng, Maomao Wang, Haiyan Li, Yongnian Qi, Qi Wang, Zhiyuan Rui

**Affiliations:** 1School of Mechanical and Electronical Engineering, Lanzhou University of Technology, Lanzhou 730050, China; frcly@163.com (R.F.); 13893591491@163.com (Y.Q.); 18350122524@163.com (Q.W.); zhiy_rui@163.com (Z.R.); 2State Key Laboratory of Advanced Processing and Recycling of Non-ferrous Metals, Lanzhou University of Technology, Lanzhou 730050, China

**Keywords:** two-phase Ti–Al alloy, void, molecular dynamics, cold deformation

## Abstract

Cold deformation behavior of polycrystalline metallic material is affected by intrinsic defects such as dislocations, voids, inclusions etc. Existing studies on $\alpha_{2}\left({\mathrm{Ti}}_{3}\mathrm{Al}\right)$ + γ(TiAl) two-phase Ti–Al alloy cover about deformation behavior mainly on macro scale. This paper focuses on the cold deformation mechanism of two-phase Ti–Al alloy at micro scale, and the role of voids in deformation process. Molecular dynamics simulations were performed to study the evolution of micro structure of material under uniaxial tension. Interaction between spherical nano voids with different size and position was also examined in the simulation. The results show that (1) In elastic stage, deformation of the two-phase is coordinated, but ${\mathrm{Ti}}_{3}\mathrm{Al}$ is more deformable; (2) In plastic stage, γ phase is the major dislocation source in two-phase alloy; (3) voids detracts the strength of the two-phase alloy, while the position of void affect the degree of this subtraction, voids located at the boundary of $\alpha_{2}$/γ phase have significant detraction to strength.

## 1. Introduction

Titanium–aluminum-based intermetallic alloys are promising high temperature structural materials because of their great corrosion resistance and strength at high temperature. Compared with nickel-based alloy, Ti–Al-based alloys have lower density, it can be used in combustion engine, turbine, and other components working at high temperatures from 500 ${}^{\circ }C$ to 900 ${}^{\circ }C$ [1]. The performance of inner combustion engine with components made of Ti-6Al-4V was improved 20% because the weight of components decreased from 30% to 40% [2]. With the advancement of non-ferrous metallurgy, Ti–Al-based alloy components have better economic efficiency than ever before. Ti–Al-based alloys have good prospects for applications in aerospace and automobile industry. Poor ductility of Ti–Al alloy at room temperature strongly affects the safety of structures like turbo of aircraft engine and combustion generator [3].

Deformation phenomena of Ti–Al alloys have been widely studied in order to overcome the problems associated with the poor ductility and damage tolerance. Much of the work has been performed on single-phase γ-TiAl, ${\mathrm{Ti}}_{3}\mathrm{Al}$ alloys, and polysynthetically twinned crystals (PTC) [4]. Experiments reveals that the dominant deformation mode in ${\mathrm{Ti}}_{3}\mathrm{Al}$ at low temperature is slip activity without any twinning, and the fracture type is cleavage [5]. Around the1990s, experimental studies on the two-phase lamellae Ti–Al alloys showed that mechanical twin and glide of dislocation are major sources of deformation [6,7,8]. Rapture failure at the macroscopic scale can be attributed to abrupt nucleation, growth and propagation of cracks, but at the microscopic scale, defects are initially formed in the casting process, such as voids and inclusions [9]. Mess research covers a wide range of factors such as alloy composition, microstructure and deformation temperature, some reports come up with the idea that two-phase Titanium–aluminum alloys with proper phase distribution and grain size exhibit better mechanical performance compared with monolithic constituents γ(TiAl) and $\alpha_{2}$(${\mathrm{Ti}}_{3}\mathrm{Al}$) alloy [10]. In situ experiments have shown that Ti–Al and ${\mathrm{Ti}}_{3}\mathrm{Al}$ in two-phase alloy exhibit different properties comparing in single-phase alloys [11]. voids can be arose due to specific volume differences induced by precipitation, different thermal expansion or shrinkage upon heating or cooling the specimen. It has been known that nucleation, growth and coalescence of voids are deemed as the primary mechanism of ductile material fracture, in which voids growth is particularly important [12]. The initiation of crack at microscopic scale is a dynamic process, which results in difficulties on study of mechanism about deformation and cracking, these defects are known playing a fundamental role in the deformation of materials. Multiscale method has been applied to study deformation behavior of polycrystal with single aluminum [13] and titanium element respectively [14]. It is necessary to carefully examine the revolution of defects and its influence on the fracture process at atomic scale.

The effect of voids is another great concern of properties and deformation mechanism of Ti–Al alloys. Therefore, it is necessary to study the deformation response of intermetallics structural materials with the consideration of microstructure evolution. Previous study on voids growth in single crystal γ-TiAl reveals that voids with high volume fraction detracts yield strength [9,15]. Evolution of voids in ductile polycrystalline was studied in nanoscale with molecular dynamics(MD) simulations, [16,17], voids inside material are sources of dislocation and affects the properties of materials differently because of differences in size and position. The deformation and fracture mechanisms in the duplex microstructure are plasticity induced grain boundary decohesion and cleavage, while those in the lamellar micro-structure are interface delamination and cracking across the lamellar [9]. It reveals that existence of voids alone may contribute to strain hardening because they are barriers to dislocation movement in ductile fcc structure metal [18]. However, few studies deal with the deformation mechanism of two-phase Ti–Al alloys and the role of voids in atomic scale. Defects are inevitable as micro-pores and loosen from casting, and in the actual work environment with radiation. A lot of work carried out on the effect of various defects on the behavior of different materials, showing that point defects may affect the properties of materials greatly. The mechanical performance of irradiated copper is affected by the interaction between irradiation and dislocation [19]. Vacancy concentration in single crystal and polycrystal Fe–40 at. Al bulk results in an increase of strength [20]. Ti–Al alloy is a type of typical brittle material, thus, it can be assumed that its properties are sensitive to the existence of void. Surface defects such as small notches can cause low and high cycle fatigue strength of the Ti–47Al–2W–0.5Si alloy [21], and the strength of single crystal γ-TiAl is also lowered by point defect [22]. The resistance of Ti–6Al–4V alloy, which was processed for typical fan blade applications, to high-cycle fatigue in the presence of foreign- object damage was reduced due to earlier crack initiation. The nucleation and subsequent near-threshold growth of crack was primarily affected by the stress concentration associated with the foreign-object damage and the presence of small cracks in the damaged zone. Due to difficulties in observing the dynamic process during deformation wit experiments, MD simulation has become an effective method to investigate micro deformation mechanism. Defects such as grain boundary, voids and segregation play significant roles in the process of fracture [23]. This paper focus on the evolution of microstructure, tending to find out its connection to cold deformation behavior of two-phase Ti–Al alloys. MD simulation including model creation and analysis method is given in Section 2, Results and discussion are in Section 3.

## 2. Molecular Dynamics Simulation

### 2.1. Atomic Potential

The interaction of particles in the material is determined by interatomic potential. Many reported simulation cases of deforming and crack propagation in metal materials were performed with embedded atomic method due to its better accuracy in metal lattice compare with F-S and L/J [24,25,26]. Embedded atom method (EAM) potential developed by Zope and Mishin [27] was used in the study. The simulation is performed with the Large-scale Atomic/Molecular Massively Parallel Simulator (LAMMPS) open-source code [28]. We did constant-pressure and constant-temperature (NPT) MD simulation at room temperature (298K). Definition of the potential is as following:(1)$$E_{total}=\sum F_{i}\left(\rho_{h,i}\right)+\frac{1}{2}\sum_{i}\sum_{j(\neq 1)}\phi_{ij}\left(R_{ij}\right)$$where $E_{total}$ is the total energy of the system, $\rho_{h,i}$, is the host electron density at atom *i* due to the remaining atoms of the system, $F_{i}\left(\rho \right)$ represents the energy for embedding atom *i* into the background electron density ρ, and $\phi_{ij}\left(R_{ij}\right)$ gives the core-core pair repulsion between atoms *i* and *j* separated by the distance $R_{ij}$. It can be noted that $F_{i}$ only depends on the element of atom *i* and $\phi_{ij}$ only depends on the elements of atoms *i* and *j*.

### 2.2. Modeling

Two phase Ti–Al alloy is composed of γ-TiAl and $\alpha_{2}$-${\mathrm{Ti}}_{3}\mathrm{Al}$, TiAl has a fcc type cell with an $L1_{0}$ structure, and ${\mathrm{Ti}}_{3}\mathrm{Al}$ has a hcp structure, these two types of initial cells are shown in Figure 1. Geometry parameters of the two type of unit cell are given by Table 1. Periodic boundary conditions (PBC) were applied along three directions, which makes polycrystal with periodic nanovoids structures. The initial size of simulation box is $L_{x}=200$ Å, $L_{y}=$180 Å, $L_{z}=210$ Å, and each model about 460,000 atoms. The model includes 6 grains, with random shape and orientation created by Voronoi method. Total simulation time is restricted to computer power, thus, strain rate in molecular dynamic simulation is much greater than physical experiments. Attempts have been done to find decent time step and strain rate for tensile test with MD simulation. If the strain rate is too high and time step is too large in the simulation, the model cannot predict the real behavior of dislocations. Previous work [22,29] reveals that a strain rate from $10^{8}\;s^{-1}$ to $10^{9}\;s^{-1}$ is appropriate for a system composed of metal atoms like Ti–Al metallic materials. thus, the strain rate of uniaxial loading was set as $5\times 10^{8}\;s^{-1}$.

### 2.3. Analysis Method

In order to identify typical defects in the deformed model, a hybrid analysis method was used with free code ovito [30]. Dislocation is visualized by DXA method, and centro-symmetry parameter (CSP) is used to differ grain boundaries, $\alpha_{2}$ phase grains and γ phase grains. The definition of CSP is as following:(2)$$P=\sum_{i=1}^{6}{\left|{\vec{R}}_{i}+{\vec{R}}_{i+6}\right|}^{2}$$where ${\vec{R}}_{i}$ and ${\vec{R}}_{i+6}$ are the vectors corresponding to the six pairs of opposite nearest neighbors in the fcc lattice. The centro-symmetry parameter (CSP) is zero for atoms in a perfect lattice. In other words, if the lattice is distorted, the value of P will not be zero. Instead, the parameter will have a value within the range corresponding to a particular defect. By removing all the perfect and surface atoms within the bulk, atoms around defected zones are visualized.

### 2.4. Model Verification

The results of the molecular dynamics simulation greatly rely on accuracy of potential and controlling parameters such as strain rate, temperature and relaxation time. Thus, we performed model verification on a simplified model and then compared the results with experiments and simulation done by others. Uniaxial tensile loading was applied to single crystal γ-TiAl bulk shown by Figure 2a, potential and relaxation parameters are the same as those which applied to the polycrystalline model. When temperature is 300 K, the sample was tensioned along [0 0 1] direction at a constant stain rate of $5\times 10^{8}\;s^{-1}$. The yield strength of the single crystal Ti–Al alloy is 9.5 GPa and that is in good agreement with work by Tang [31] in consideration of size effect, and the snapshoot of the fracture surface is shown by Figure 2, and compared with experiment results by Cao [32] shown in Figure 2c. It should be noticed that the image from SEM is in a greater scale than our simulation cell. However, the cracked surface in atomic model exhibits two major characteristics of TiAl alloy. First, the fracture type of TiAl at room temperate is typical brittle failure. Second, the fracture surface of atomic model in Figure 2b is typical cleavage surface. This verification case has proven that the EAM potential is of high efficiency and accurate enough, and the simulation work flow is valid to predict failure behavior of the system composed of Ti–Al.

## 3. Results and Discussion

Deformation process of the model without void is shown in Figure 3, the strain of model at 10 key points are given by Table 2. The strength of the model without void is 5.3 GPa. According to stress response under constant rate of strain rate, the whole tensile process can be divided into four stages: stage-I: elastic stage, from ϵ=0 to ϵ=0.092, including key point 1, stage-II: yield stage, ranging from ϵ=0.092 to ϵ=0.101, including key points 2 to 6, stage-III: cracking stage, ranging from ϵ=0.101 to ϵ=0.112, including key point 7 to 10, stage-IV: fracture stage. Following discussion concentrates on deformation phenomena that rely on the elastro-plastic code formation of the γ and $\alpha_{2}$ phases and on the particular point defect situation occurred in two-phase alloys.

### 3.1. Deformation Mechanism of Two Phase Ti–Al Alloy

During the elastic stage, the deformation of $\alpha_{2}$ phase and γ phase grains is compatible, however the properties of two phases are different. The elastic constants given by experiments [33,34] are listed in Table 3. $C_{11}^{\alpha_{2}}$ is 176, which is smaller than $C_{11}^{\gamma }$, other five parameters but $C_{33}$ of ${\mathrm{Ti}}_{3}\mathrm{Al}$ are also smaller than Ti–Al. In order to quantify the difference of the two-phase deformation gradient was calculated by formulation:(3)$$F^{e}=\frac{\partial x}{\partial X}=\mathrm{Grad}x,\;F_{iJ}=\frac{\partial x_{i}}{\partial X_{J}}$$we computes the atomic level elastic strain and deformation gradient tensors in crystalline systems with the numerical method proposed by [11]. When global strain is 0.015, local elastic strain $F_{xx}$ of the model is measured along the line cross center of the model and shown by Figure 4. In te figure, $\alpha_{2}$ phase located in the central part of the axial, and other parts on the axial are γ phase. It should be noticed that atomic level elastic strain effected by random thermal displacements of atoms, thus, fluctuation value exists in Figure 4. The local strain of $\alpha_{2}$ phase grain is greater than γ phase grain when global strain of the model is 0.015. From the value of local strain, we can conclude that γ phase less deformable than $\alpha_{2}$ phase during elastic stage.

Snapshots of atoms configuration at the three of ten key points are shown by Figure 5. Atoms with ordered configuration of γ phase grains have been removed in Figure 5a,b, $\alpha_{2}$ phase and defects inside grains have been left. Similarly, γ phase grain have been removed in Figure 5c,d, the defects of $\alpha_{2}$ phase have been left. The results show that, at stage I, the structure of material is under typical elastic deformation, the size of simulation box enlarged due to the loading. In this stage, deformation of the two-phase are compatible. Emission of dislocation and evolution of defects initiated at the end of the stage I. A great number of dislocation emitted inside γ phase at earlier part of stage II, however, the dislocation inside $\alpha_{2}$ phase was emitted at the end of stage II is shown by Figure 5c.

Experiments have shown that the velocity of dislocation motion is sensitive to stress and temperature [35]. The brittleness of two-phase Ti–Al alloy attributed to the poor mobility of dislocation at room temperature. Velocity of a screw dislocation can be estimated by Escaig’s elastic model [36], it can be written as:(4)$$v=v_{0}\;e^{-\Delta H(\tau^{*})/kT}$$where the prefactor $v_{0}$ gives the velocity that would be obtained for each potential mobility, *L* represents the free length of screw character of dislocation, $\Delta H(\tau^{*})$ is activation enthalpy determined by loading conditions. The effect of temperature on the mobility can be evaluated under different loading conditions, thus, we chose $\tau_{1}^{*}>\tau_{2}^{*}>\tau_{3}^{*}$ in Formulation (Equation 4), normalized velocity of dislocation motion is shown by Figure 6. The velocity of dislocation movement rise along with increase of stress, the mobility of dislocation is relatively poor at 298 K. Due to motion of dislocations is inactive inside $\alpha_{2}$ phase, typical strengthening mechanism closely related to the interaction between grain boundary and dislocation can not be dramatic at room temperature, piling up of dislocations is not observed in this simulation, which is shown in Figure 5.

Length of different types of dislocations are given at four stages in Figure 7. Dislocations are recognized and identified using the dislocation analysis (DXA) of ovito. The length of Shockley dislocation with Burgers vector 1/6[112] increased sharply during yield stage of the deformation process in Figure 7, but the length of perfect dislocation fluctuated at a low level. Perfect dislocations split into Shockley partial dislocations, this decomposition process can be given by Formulations (Equation 5) and (Equation 6). The structure spontaneously transformed into intrinsic stacking fault (ISF), which can be observed in Figure 5a.(5)$$1/2\left[\;0\;1\;1\;\right]\to 1/6\left[\;1\;1\;2\;\right]+1/6\left[\;\bar{1}\;2\;1\;\right]$$
(6)$$1/2\left[\;1\;0\;\bar{1}\;\right]\to 1/6\left[\;1\;1\;\bar{2}\;\right]+1/6\left[\;2\;\bar{1}\;\bar{1}\;\right]$$Interaction between dislocations with Burgers vector of 1/6[112] and $1/6[11\bar{2}]$ follows the decomposition of perfect dislocation. The process produced leading dislocation and trailing dislocation, when the two leading Shockley partials combined, they form Lomer–Cottrell junction, a separate dislocation. It’s sessile in the slip plane, playing a role of barrier against other dislocations in the plane. This process is given by:(7)$$1/6\left[\;1\;1\;2\;\right]+1/6\left[\;1\;1\;\bar{2}\;\right]\to 1/3\left[\;1\;1\;0\;\right]$$According to the results calculated by first principle, the stacking fault energy of Ti–Al is much smaller than ${\mathrm{Ti}}_{3}\mathrm{Al}$, as a consequence, calculated shear strength of Ti–Al and ${\mathrm{Ti}}_{3}\mathrm{Al}$ is 4.1 GPa and 2.86 GPa along their easy slip plane respectively [37]. This calculation reveals that tensile behavior in single crystal is mainly controlled by the type of the structure. However, in two-phase Ti–Al alloy, deformation of $\alpha_{2}$ phase is earlier than γ phase during yield stage, thus, local displacement of two phases are incompatible during yield stage. Mobility of dislocation is affected by structure of crystal, loading condition and temperature, and $\alpha_{2}$ phase is made up of hcp structured grain, grains with this type of structure possess less slip system compared with grain with fcc structure [38], thus, $\alpha_{2}$ phase is difficult do deform under uniaxial tensile loading.

TEM examinations performed on tensile-tested lamellar alloys have revealed that the limited plasticity of $\alpha_{2}$ phase is carried by local slip of dislocations with the Burgers vector $1/3[11\bar{2}0]$ prism planes shown by Figure 5b, which is by far the easiest slip system in $\alpha_{2}$ single crystals. Dislocation neighboring interfaces often needs to be transformed. For example, an ordinary 1/2 [110] dislocation gliding in one γ grain has to be converted into [101] super dislocation when the double Burgers vectors gliding in an adjacent γ grain. At the $\alpha_{2}/\gamma$ interface, dislocations existing in $D0_{19}$ structure transform into dislocations, which is consistent with the $L1_{0}$ structure. These core transformations are associated with the change of the dislocation line energy because of the differences of the length and the shear module. Pyramidal slip of the $\alpha_{2}$ phase is required when slip is forced to cross $\alpha_{2}$ phase, this process needs an extremely high shear stress. Two components ($\alpha_{2}$ and γ phase) exhibit different properties in two-phase alloy comparing they are single-phase alloy respectively. γ phase is the major source of dislocation in two-phase alloy, that is in good agreement with experiments [8,11]. When global strain is greater than 0.07, the reason why Ti3Al exhibits less plastic deformation is the absence of twinning in Ti3Al under tensile loading, which have been observed by experiments [5]. The deformation ${\mathrm{Ti}}_{3}\mathrm{Al}$ is mainly constrained by loading conditions, as compression loading is applied, Ti3Al activates more moveable dislocation, thus, exhibits better elasticity [39].

### 3.2. The Effect of Void on the Strength of Material

In Figure 8, voids with different size of 2 Å, 5 Å, 10 Å and 15 Å were placed at $\alpha_{2}$/γ phase boundary, inside γ phase respectively. The strength of materials with void in different size and at different position is shown in Figure 9. The existence of void have little impact on the elastic properties of the material, and the model without void has the largest strength 5.3 GPa, the yield stress of model with void is smaller. Void at $\alpha_{2}$/γ phase boundary detracts the strength of material most, and the void inside $\alpha_{2}$ phase have less impact on the strength. Conventional definition of strength of materials with geometry subtraction was applied to the model, and theoretical strength of the models was calculated by(8)$$\sigma^{*}=\sigma_{0}\cdot \left(A^{*}/A_{0}\right)$$where $\sigma_{0}$ is the strength of model without voids of 5.3 GPa, and $A_{0}$ represents initial section area, $A^{*}$ is effective section area in consideration of the area detraction by void. In classic theory, the relationship between void size and strength of the model is linear. However, simulation results reveals that two-phase Ti–Al alloy is sensitive to defect of void. Comparing the strength determined from molecular dynamics simulation and the results calculated with Formulation (Equation 8), it can be seen that the main factor that affects the strength of materials is local behavior of the materials, thus, evolution of defects have dramatic influence on the yield and fracture process of the materials.

It has been observed in Figure 10 that voids detract the strength of materials. The max stress of the simulation cell decreases as the volume of voids increases. It can be seen from Figure 9 that there is a critical value of void radius about 15 Å, the void greater than 15 Å causes serious detraction of strength of material. The decreased rate of loading area is smaller compared with the detraction of strength, so it can be assumed that the yield behavior and strength is more related with local behavior of grain boundaries and void. Grain and phase boundaries are obstacles to deformation process, thus, the stability of boundaries has a great impact on the strength of materials. Interaction between grain boundary and void determines the fracture mode of Ti–Al alloy.

### 3.3. Evolution of Spherical Void

The role of void can be concluded as two main parts: source of dislocations and obstacles to dislocations, which is dominant depends on the location of void. Void inside $\alpha_{2}$ phase plays a role of inclusion, in other words, this type of void possess similar properties as second phase particles.

Void inside $\alpha_{2}$ phase induced considerable misfit-stress fields and thus, can influence material properties. Such stress fields surrounding the second-phase particles can be due to misfit between $\alpha_{2}$ phase atoms and surface atoms surrounding the void. That is a possible favorable effect of second-phase particles that void contribute to the enhancement of mechanical strength. The strengthening effect of the particle inside single-phase Ti–Al alloy has been verified by experiment [40]. Considering yielding of a material as related to glide of dislocations, any mechanism obstructing dislocation glide improves the mechanical strength. The defect evolution neighboring the void during yield stage is shown in Figure 11a; the role of void is similar to second-phase particles. It servers as obstacles for dislocation migration which is shown by Figure 11b: the stress fields surrounding the second-phase particles block migrating dislocation, the void particle acts as pinning point. The dislocation can pass two pinning points under shear stress, and the critic value depends on the distance between obstacles, it is be given by [18]:(9)$$\tau_{0}=Gb/d$$where *d* represents the distance between two obstacles A and B in Figure 11, reflects the dependence of the critical shear stress $\tau_{0}$ on the second-phase particle density and distribution. This mechanism for hardening is designated as the Orowan process with $\tau_{0}$ as the Orowan shear stress. As a result of the Orowan process, upon passage of the pinning points by a series of gliding dislocations, a system of concentric loops is formed around the second-phase particles. Consequently, the effective average distance between the second phase particles has decreased to *d* which implies a necessary increase of the value of critical shear stress required for continuation of dislocation glide. The width of a Burgers vector, will be generated at both sides of a crystal along the direction of burgers vector after dislocation traversing the entire crystal, as is shown in the third subfigure of Figure 11b.

Effect of voids at different positions on the crack mode of the material are shown by Figure 12a. The model with void fractured cross the grain, and the model with void at $\alpha_{2}/\gamma$ boundary resulted a large crack along the boundary under tensile loading. The difference of crack mode can attribute to the position of void and the evolution of the defect. Nucleation and emission of Shockley partial dislocation affect the evolution of the void at boundary greatly due to the anisotropy of two phases on the both sides of the phase boundary. The void weakens the strength of the grain it locates, $\alpha_{2}$ phase grain is easy to deformed under shear stress, thus, that caused the crack initiated inside grain and finally caused fracture across the grain. Deformation mechanisms of the model with void at boundary can be more complex due to the interaction between void and phase boundary. In the elastic stage, the system composed of $\alpha_{2}$ phase, γ phase and void is in equilibrium, internal stress are balanced under compatible deformation as is shown by the first subfigure of Figure 12b. In the middle moment of yield stage, partial dislocation with Burgers vector $1/6[\bar{2}\;\bar{1}\;1]$ emitted from surface of void in Figure 12b-2. The micro crack initialed from the zone neighbor the interface between void and boundary, that quickly caused abrupt decohension of phase boundary. That accounts for the reason that material with void at $\alpha_{2}/\gamma$ phase boundary is weaker than other cases.

## 4. Conclusions

In this paper, deformation behavior of two-phase Ti–Al alloy under tensile loading was simulated with MD method. The mechanism of deformation was investigated under atomic scale, and the effect of void on the properties of two-phase Ti–Al alloy was also studied. The conclusions are as follows:

(1) Deformation behavior of two types of grain inside $\alpha_{2}+\gamma$ two-phase Ti–Al alloy are different due to their crystal structure. $\alpha_{2}$ phase ${\mathrm{Ti}}_{3}\mathrm{Al}$ grain is easy to be deformed during elastic stage.

(2) During the yield stage, the majority of dislocations in two-phase Ti–Al alloy activated inside γ phase, $\alpha_{2}$ phase is harder to slip due to its hcp structure, this inhomogeneity results in cracks at interface of the two phases.

(3) Effect of void on the strength of the two-phase alloy is sensitive to the location of void. Void inside grain has detraction to the strength of material because the strengthening mechanism is similar to second-phase particles. Void at $\alpha_{2}/\gamma$ boundary is the most risky situation for the two-phase alloy because of fast fractures along boundary.

## Figures and Tables

**Figure 1 materials-12-00184-f001:**
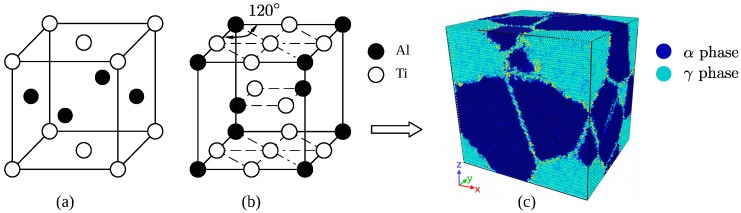
Unit cell of TiAl (**a**), ${\mathrm{Ti}}_{3}\mathrm{Al}$ (**b**) and two-phase Ti–Al model (**c**).

**Figure 2 materials-12-00184-f002:**
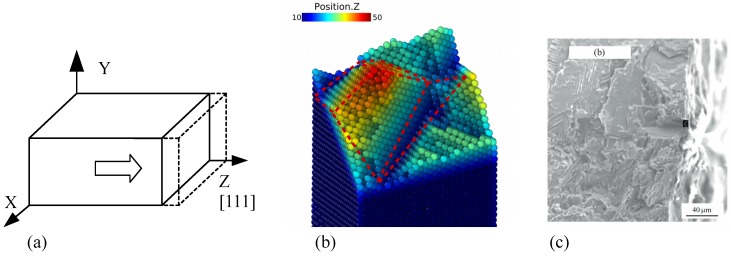
(**a**) Verifying model: TiAl single crystal bulk; (**b**) Cracked surface of atomic model; (**c**) Cracked surface from experiment [32].

**Figure 3 materials-12-00184-f003:**
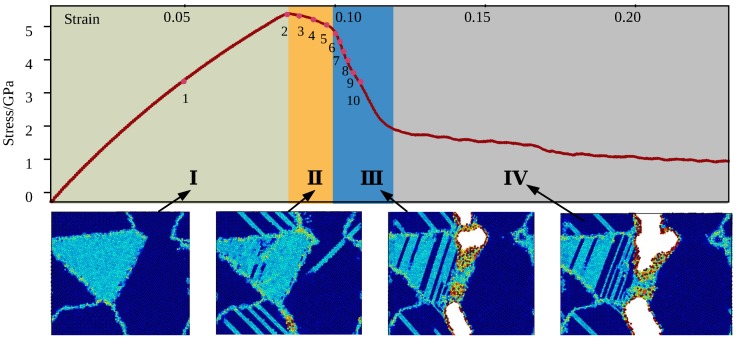
Deformation process of the model without void.

**Figure 4 materials-12-00184-f004:**
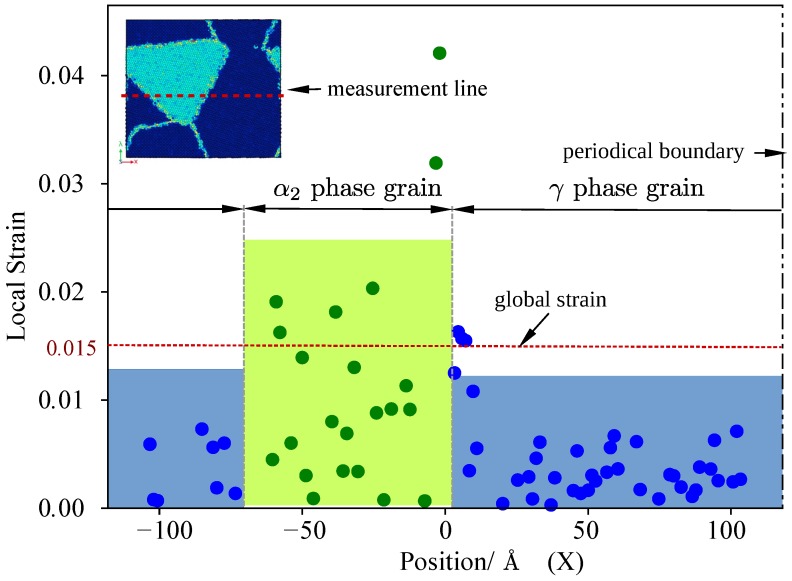
Local strain along tensile direction.

**Figure 5 materials-12-00184-f005:**
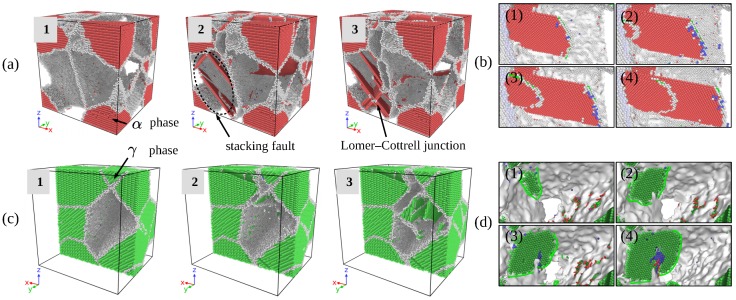
Microstructure evolution inside γ phase (**a**,**b**), $\alpha_{2}$ phase (**c**,**d**) during yield stage.

**Figure 6 materials-12-00184-f006:**
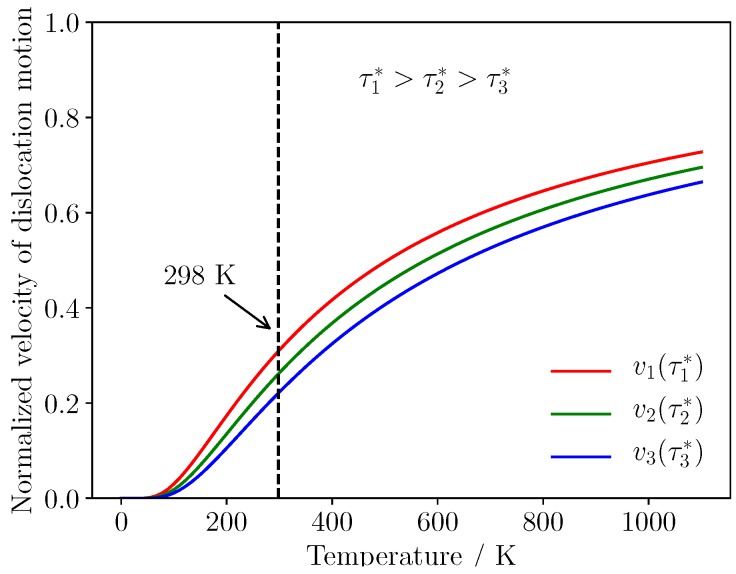
Normalized velocity of dislocation motion under different loading conditions.

**Figure 7 materials-12-00184-f007:**
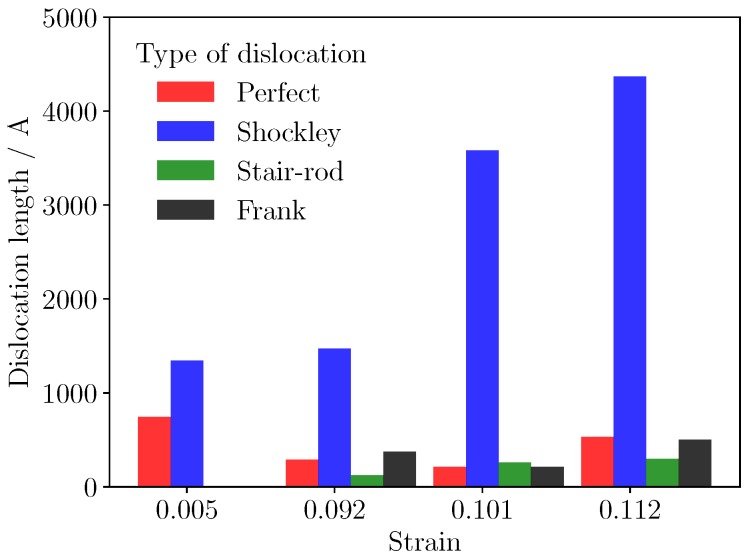
Summary of didderent types of dislocation under different strain.

**Figure 8 materials-12-00184-f008:**
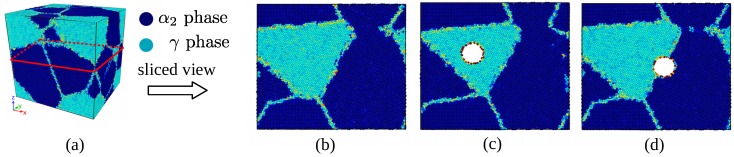
Model of the simulation (**a**) and sliced view of the model with no void (**b**), inside $\alpha_{2}$ phase (**c**) at $\alpha_{2}/\gamma$ phase boundary (**d**).

**Figure 9 materials-12-00184-f009:**
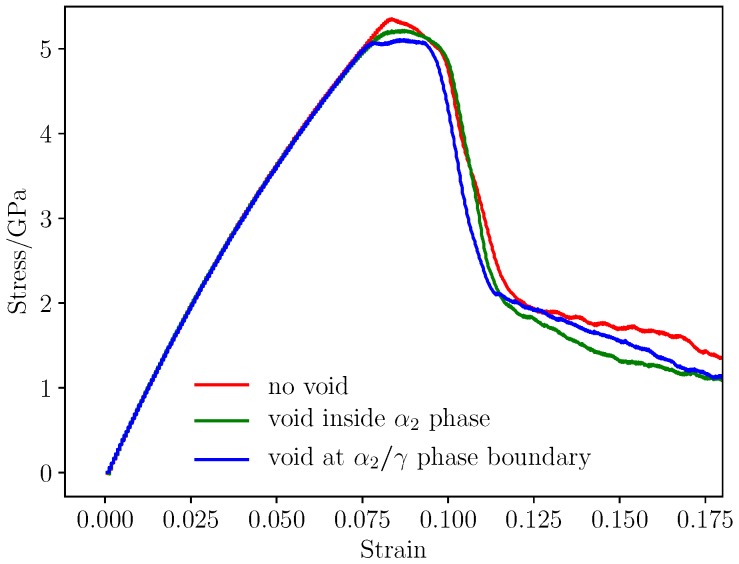
Stress-strain response of the materials under tensile loading.

**Figure 10 materials-12-00184-f010:**
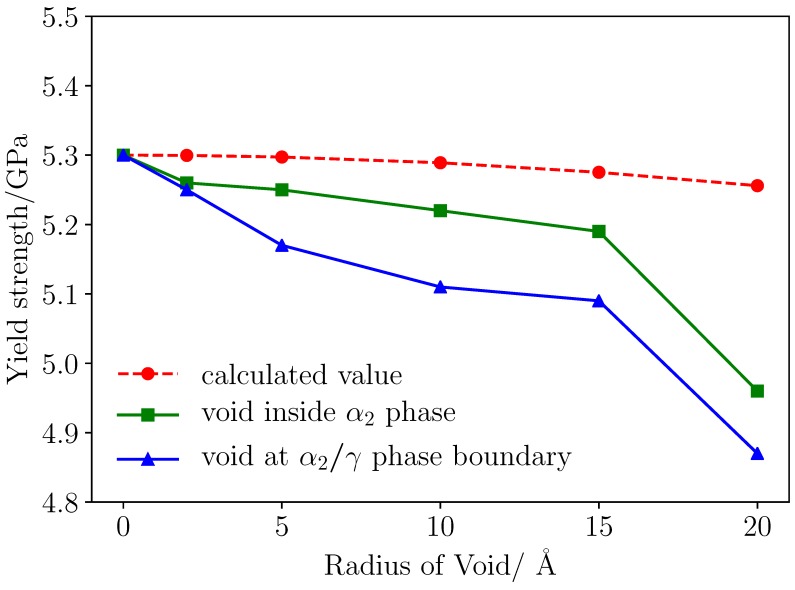
Yield strength of the materials with different size void.

**Figure 11 materials-12-00184-f011:**
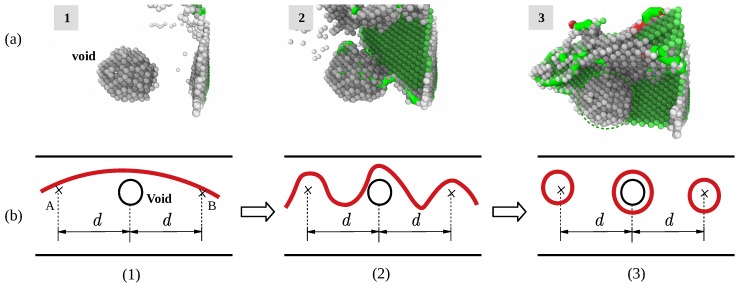
(**a**) Orowan process occurred inside $\alpha_{2}$ phase grain (removed atoms with regular arrangement). (**b**) Schematic of Orowan process.

**Figure 12 materials-12-00184-f012:**
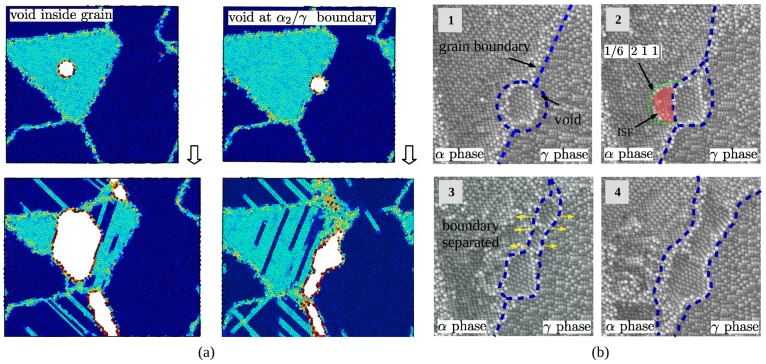
(**a**) Fracture mode of two types of mode; (**b**) Evolution of void at $\alpha_{2}/\gamma$ phase boundary.

**Table 1 materials-12-00184-t001:** Parameters of nanocrystalline.

Phase	Space Group	Designation	Parameters
$\alpha_{2}$-${\mathrm{Ti}}_{3}\mathrm{Al}$	$P6_{3}/\mathrm{mmc}$	$0_{19}$	*a* = 0.5765
			*c* = 0.46833
γ-TiAl	tP4	$L1_{0}$	*a* = 0.3997
			*c* = 0.4062

**Table 2 materials-12-00184-t002:** Key points during tensile process.

Number	1	2	3	4	5	6	7	8	9	10
**Stage**	I	I	II	II	II	II	III	III	III	III
**Strain**	0.05	0.088	0.092	0.096	0.099	0.101	0.104	0.107	0.110	0.112

**Table 3 materials-12-00184-t003:** Elastic constants of TiAl and ${\mathrm{Ti}}_{3}\mathrm{Al}$.

	$C_{11}$	$C_{12}$	$C_{13}$	$C_{33}$	$C_{44}$	$C_{66}$
TiAl [33]	186	72	74	176	101	77
${\mathrm{Ti}}_{3}\mathrm{Al}$ [34]	176	87.8	61.2	218.7	91.9	62.4

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
