# Peer review of "Micromechanism of Cold Deformation of Two-Phase Polycrystalline Ti–Al Alloy with Void"

_materials, 2019, doi:10.3390/ma12010184_

Round 1
Author Response
Response to Reviewer Comments
Thanks for your time, and we did following changes on our script according to your feedback:
Point 1: Introduction can be improved
Response 1:
We added 4 citation in INTRODUCTION section, these articles focus on deformation mechanism of two phase Ti-Al alloy, ranging from 1980 to 2017.
1. Sastry, S.M.L.; Lipsitt, H.A. Plastic deformation of TiAl and Ti3Al. Proceedings of the 4th International Conference on Titanium, 1980, pp. 1231–1243.
2. Farenc, S.; Coujou, A.; Couret, A. An in situ study of twin propagation in TiAl. Philosophical Magazine A: Physics of Condensed Matter, Structure, Defects and Mechanical Properties 1993, 67, 127–142. doi:10.1080/01418619308207147.
3. Appel, F.; Wagner, R. Microstructure and deformation of two-phase g-titanium aluminides. Materials Science and Engineering R: Reports 1998, 22, 187–268. doi:10.1016/S0927-796X(97)00018-1.
4. Hempel, N.; Bunn, J.R.; Nitschke-Pagel, T.; Payzant, E.A.; Dilger, K. Study on the residual stress relaxation in girth-welded steel pipes under bending load using diffraction methods. Materials Science and Engineering A 2017, 688, 289–300. doi:10.1016/j.msea.2017.02.005.
Point 2: Research design can be improved
Response 2:
Model verification was added into section 2
In order to prove the accuracy of molecular simulation, we added Model Verification (subsection 2.4) into section 2. We designed a simplified atomic model for bulk single crystal TiAl alloy, and the results were compared with experimental works done by our institution previously.
5. Chen, J.H.; Cao, R. Chapter 9 - Brittle Fracture of TiAl Alloys and NiTi Memory Alloys. In Micromechanism of Cleavage Fracture of Metals; Chen, J.H.; Cao, R., Eds.; Butterworth-Heinemann: Boston, 2015; pp. 365–443.doi:https://doi.org/10.1016/B978-0-12-800765-5.00009-5.
6. Tang, T.; Kim, S.; Horstemeyer, M.F. Molecular dynamics simulations of void growth and coalescence in single crystal magnesium. Acta materialia 2010, 58, 4742–4759. doi:10.1016/j.actamat.2010.05.011
The verifying model is shown in image (In attached PDF file):
The verifying model reveals that:
1. Atomic potential is accurate enough to predict the strength of the Ti-Al alloy.
2. Cracked surface of atomic simulation is typical cleavage surface, that is in good agreement with experimental work don by Cao[1]. It should noticed that the SEM image is in a greater scale than atomic simulation, however, the atomic model exhibits dramatic brittleness and typical characteristic of rapture fracture, thus we think the atomic simulation is descent to predict failure mode of TiAl alloy.
Point 3: Methods description can be improved
Response 3:
The strain rate of our simulation is 5e8/s-1, the value is much higher than real physical tensile experiment. The reason why the value of strain rate is so high is the balance of compute power and physical reality. In other words, on the one hand, if the strain rate is too low, computing time would be too long. Total simulation time at atomic scale is below 10ns according to reported works, thus a higher strain rate is needed in consideration of efficiency. On the other hand, high strain rate have effect on evolution of dislocation, previous work have reveals that, if the strain rate is greater than 1e9/s-1, atomic simulation will be far from reality. Following research articles focus on the effect of strian rate in atomic simulation:
7. Brimmo, A.T.; Hassan, M.I.; Shatilla, Y. Transient heat transfer computational model for the stopped aluminium reduction pot - Cooling techniques evaluation. Applied Thermal Engineering 2014, 73, 114–125.doi:10.1016/j.jmst.2015.12.001.
8. Zhu, T.; Li, J.; Samanta, A.; Leach, A.; Gall, K. Temperature and strain-rate dependence of surfacedislocation nucleation. Physical Review Letters 2008, 100, 025502. doi:10.1103/PhysRevLett.100.025502.
Point 4: Results presentation can be improved
Response 4:
Discussion about elastic deformation was added to the RESULTS AND DISCUSSION(section 4).
Point 5: Conclusion can be improved
Response 5:
We add a point conclusion in section:Deformation behavior of two types of grain inside two phase Ti-Al alloy are different due to their crystal structure. Ti3Al grain is easy to be deformed during elastic stage.

Reviewer 2 Report
This paper studies the effect of the void on the cold deformation of two phases Ti-Al alloy. The research is interesting, however to be accepted for publication the following comments need to be addressed
1-English Minor spell check is required.
2- The author should support his clam by some experimental work if it possible. Uniaxial tensile test at one simulation condition then, the microstructure analysis after fracture using TEM microscope should be involved.
3- In Model Creation of Crystalline section, please check the value of the strain rate.
Author Response
Response to Reviewer Comments
Thanks for your time and valuable advice, we did following changes on our script according to your feedback, and all the changes have marked on the new edition script file: diff.pdf
Point 1: English Minor spell check is required
Response 1:
Some of spelling mistakes and grammar problems have been corrected,
Point 2: The author should support his clam by some experimental work if it possible. Uniaxial tensile test at one simulation condition then, the microstructure analysis after fracture using TEM microscope should be involved.
Response 2:
Honestly, experimental test for our case faces difficulties because of observation methods.
In order to prove the accuracy of molecular simulation, we added Model Verification (subsection 2.4) into section 2. We designed a simplified atomic model for bulk single crystal TiAl alloy, and the results were compared with experimental works done by our institution previously.
1. Chen, J.H.; Cao, R. Chapter 9 - Brittle Fracture of TiAl Alloys and NiTi Memory Alloys. In Micromechanism of Cleavage Fracture of Metals; Chen, J.H.; Cao, R., Eds.; Butterworth-Heinemann: Boston, 2015; pp. 365–443.doi:https://doi.org/10.1016/B978-0-12-800765-5.00009-5.
2. Tang, T.; Kim, S.; Horstemeyer, M.F. Molecular dynamics simulations of void growth and coalescence in single crystal magnesium. Acta materialia 2010, 58, 4742–4759. doi:10.1016/j.actamat.2010.05.011
The verifying model is shown in image (In PDF file):
The verifying model reveals that:
1. Atomic potential is accurate enough to predict the strength of the Ti-Al alloy.
2. Cracked surface of atomic simulation is typical cleavage surface, that is in good agreement with experimental work don by Cao[1]. It should noticed that the SEM image is in a greater scale than atomic simulation, however, the atomic model exhibits dramatic brittleness and typical characteristic of rapture fracture, thus we think the atomic simulation is descent to predict failure mode of TiAl alloy.
Point 3: In Model Creation of Crystalline section, please check the value of the strain rate
Response 3:
The strain rate of our simulation is 5e8/s-1, the value is much higher than real physical tensile experiment. The reason why the value of strain rate is so high is the balance of compute power and physical reality. In other words, on the one hand, if the strain rate is too low, computing time would be too long. Total simulation time at atomic scale is below 10ns according to reported works, thus a higher strain rate is needed in consideration of efficiency. On the other hand, high strain rate have effect on evolution of dislocation, previous work have reveals that, if the strain rate is greater than 1e9/s-1, atomic simulation will be far from reality. Following research articles focus on the effect of strian rate in atomic simulation:
3. Brimmo, A.T.; Hassan, M.I.; Shatilla, Y. Transient heat transfer computational model for the stopped aluminium reduction pot - Cooling techniques evaluation. Applied Thermal Engineering 2014, 73, 114–125.doi:10.1016/j.jmst.2015.12.001.
4. Zhu, T.; Li, J.; Samanta, A.; Leach, A.; Gall, K. Temperature and strain-rate dependence of surfacedislocation nucleation. Physical Review Letters 2008, 100, 025502. doi:10.1103/PhysRevLett.100.025502.
[4] suggests that a strain rate from 108/s-1 to109/s-1is appropriate for fcc structure crystal like Ti-Al metallic materials, thus we choose 5*108/s-1
Description about the selection of strain rate was added in MOLECULAR DYNAMICS SIMULATION(section2).
Other improvements:
1. We added 4 citation in INTRODUCTION section, these articles focus on deformation mechanism of two phase Ti-Al alloy, ranging from 1980 to 2017.
5. Sastry, S.M.L.; Lipsitt, H.A. Plastic deformation of TiAl and Ti3Al. Proceedings of the 4th International Conference on Titanium, 1980, pp. 1231–1243.
6. Farenc, S.; Coujou, A.; Couret, A. An in situ study of twin propagation in TiAl. Philosophical Magazine A: Physics of Condensed Matter, Structure, Defects and Mechanical Properties 1993, 67, 127–142. doi:10.1080/01418619308207147.
7. Appel, F.; Wagner, R. Microstructure and deformation of two-phase g-titanium aluminides. Materials Science and Engineering R: Reports 1998, 22, 187–268. doi:10.1016/S0927-796X(97)00018-1.
8. Hempel, N.; Bunn, J.R.; Nitschke-Pagel, T.; Payzant, E.A.; Dilger, K. Study on the residual stress relaxation in girth-welded steel pipes under bending load using diffraction methods. Materials Science and Engineering A 2017, 688, 289–300. doi:10.1016/j.msea.2017.02.005.
2. We add a point conclusion in section:Deformation behavior of two types of grain inside two phase Ti-Al alloy are different due to their crystal structure. Ti3Al grain is easy to be deformed during elastic stage.

Reviewer 3 Report
The paper needs an improvement of the language. There are many mistypes (e.g. nickle, pron, Gpa) and incorrect constructions of sentences (e.g. Model Creation of Crystalline). Checking of the document by native / fluent English speaker is highly recommended.
The model seems to be correct and all conclusions seem to be well supported by the calculations. However, one of the main results, which shows that the gamma-TiAl phase is more deformable than Ti3Al, is not in agreement with the previous results of experiments of other authors (e.g.
https://cdn.ymaws.com/titanium.org/resource/resmgr/ZZ-WTCP1980-VOL2/1980_Vol_2.-5-Plastic_Deform.pdf). In reality, gamma phase is less deformable than Ti3Al. I recommend to include to the review also the results of experimental studies of this material and to reconsider the results in order to fit to the reality.
Author Response
Response to Reviewer Comments
Thanks for your time and valuable advice, we did following changes according to your feedback, and all the changes have marked o n the new edition script file: diff.pdf
Point 1: The paper needs an improvement of the language. ...
Response 1:
Some of spelling mistakes and grammar problems have been corrected,
Point 2: The model seems to be correct and all conclusions seem to be well supported by the calculations. However, one of the main results, which shows that the gamma-TiAl phase is more deformable than Ti3Al, is not in agreement with the previous results of experiments of other authors ...
Response 2:
Honestly, our discussion about deformation was not adequate, thus we added further discussion about elastic deformation before discussing plastic deformation, and wrong expression have modified.
(1) During elastic stage, Ti3Al is more deformable than TiAl, here is experimental value of elastic constants of TiAl and Ti3Al:
[Image and Table are in attached PDF File]
We get deformation gradient along tensile direction(Fexx) in the atomic model under 1.5% global strain(Fig. 4). This results is agreement with experiment results, we should notice that the fluctuation of value of strain is mainly because of thermal induced atom movement.
(2) Situation getting more complex during plastic stage, because two components(alpha and gamma phase) exhibit different properties in two phase alloy comparing they are single phase alloy respectively. Gamma phase is the major source of dislocation in two phase alloy, that is in good agreement with experiments [2,3]. When global strain is greater than 5%, the reason why Ti3Al exhibits less plastic deformation is the absence of twinning in Ti3Al under tensile loading, which have been observed by experiments [1]. The deformation Ti3Al is mainly constrained by loading conditions, as compression loading is applied, Ti3Al activates more moveable dislocation, thus exhibits better elasticity[3]
In a conclusion, Ti3Al is more deformable comparing with TiAl during elastic stage, and TiAl is the major source of dislocation in two phase alloy, thus induced more plastic deformation than Ti3Al[4].
1. Sastry, S.M.L.; Lipsitt, H.A. Plastic deformation of TiAl and Ti3Al. Proceedings of the 4th International Conference on Titanium, 1980, pp. 1231–1243.
2. Farenc, S.; Coujou, A.; Couret, A. An in situ study of twin propagation in TiAl. Philosophical Magazine A: Physics of Condensed Matter, Structure, Defects and Mechanical Properties 1993, 67, 127–142. doi:10.1080/01418619308207147.
3. Appel, F.; Wagner, R. Microstructure and deformation of two-phase g-titanium aluminides. Materials Science and Engineering R: Reports 1998, 22, 187–268. doi:10.1016/S0927-796X(97)00018-1.
4. habil. Fritz Appel, D.; Paul, D.J.D.H.; Oehring, D.M.; Microstructures, L. Deformation Behavior of Two-Phase a2(Ti3Al) + g(TiAl) Alloys. Gamma Titanium Aluminide Alloys 2011, 2, 125–248. doi:10.1002/9783527636204.ch6.
Other improvements
1. Model verification was added into section 2
In order to prove the accuracy of molecular simulation, we added Model Verification (subsection 2.4) into section 2. We designed a simplified atomic model for bulk single crystal TiAl alloy, and the results were compared with experimental works done by our institution previously.
5. Chen, J.H.; Cao, R. Chapter 9 - Brittle Fracture of TiAl Alloys and NiTi Memory Alloys. In Micromechanism of Cleavage Fracture of Metals; Chen, J.H.; Cao, R., Eds.; Butterworth-Heinemann: Boston, 2015; pp. 365–443.doi:https://doi.org/10.1016/B978-0-12-800765-5.00009-5.
6. Tang, T.; Kim, S.; Horstemeyer, M.F. Molecular dynamics simulations of void growth and coalescence in single crystal magnesium. Acta materialia 2010, 58, 4742–4759. doi:10.1016/j.actamat.2010.05.011
The verifying model is shown in following image (from modified script):
[Image and Table are in attached PDF File]
The verifying model reveals that:
1. Atomic potential is accurate enough to predict the strength of the Ti-Al alloy.
2. Cracked surface of atomic simulation is typical cleavage surface, that is in good agreement with experimental work don by Cao[1]. It should noticed that the SEM image is in a greater scale than atomic simulation, however, the atomic model exhibits dramatic brittleness and typical characteristic of rapture fracture, thus we think the atomic simulation is descent to predict failure mode of TiAl alloy.
2.Detailed description about selection of strain rate
The strain rate of our simulation is 5e8/s-1, the value is much higher than real physical tensile experiment. The reason why the value of strain rate is so high is the balance of compute power and physical reality. In other words, on the one hand, if the strain rate is too low, computing time would be too long. Total simulation time at atomic scale is below 10ns according to reported works, thus a higher strain rate is needed in consideration of efficiency. On the other hand, high strain rate have effect on evolution of dislocation, previous work have reveals that, if the strain rate is greater than 1e9/s-1, atomic simulation will be far from reality. Following research articles focus on the effect of strian rate in atomic simulation:
7. Brimmo, A.T.; Hassan, M.I.; Shatilla, Y. Transient heat transfer computational model for the stopped aluminium reduction pot - Cooling techniques evaluation. Applied Thermal Engineering 2014, 73, 114–125.doi:10.1016/j.jmst.2015.12.001.
8. Zhu, T.; Li, J.; Samanta, A.; Leach, A.; Gall, K. Temperature and strain-rate dependence of surfacedislocation nucleation. Physical Review Letters 2008, 100, 025502. doi:10.1103/PhysRevLett.100.025502.

Round 2
Reviewer 1 Report
The manuscript has been adequately improved in light of the original review and now warrants publication in Materials.
Kind regards!
Reviewer 3 Report
The paper has been considerably improved. Now it reflects the reality and explains well what was obseved during experiments by other authors.